# Brain Regional Identity and Cell Type Specificity Landscape of Human Cortical Organoid Models

**DOI:** 10.3390/ijms232113159

**Published:** 2022-10-29

**Authors:** Manuela Magni, Beatrice Bossi, Paola Conforti, Maura Galimberti, Fabio Dezi, Tiziana Lischetti, Xiaoling He, Roger A. Barker, Chiara Zuccato, Ira Espuny-Camacho, Elena Cattaneo

**Affiliations:** 1Laboratory of Stem Cell Biology and Pharmacology of Neurodegenerative Diseases, Department of Biosciences, University of Milan, 20122 Milan, Italy; 2Istituto Nazionale Genetica Molecolare (INGM), 20122 Milan, Italy; 3WT-MRC Cambridge Stem Cell Institute, Department of Clinical Neuroscience, University of Cambridge, Cambridge CB2 0AW, UK; 4GIGA-Stem Cells, Interdisciplinary Cluster for Applied Genoproteomics (GIGA-R), University of Liège, C.H.U. Sart Tilman, 4000 Liège, Belgium

**Keywords:** hPSC-derived 3D brain organoid models, default and directed differentiation protocols, brain regional identity, neuronal maturation, cell type specificity

## Abstract

In vitro models of corticogenesis from pluripotent stem cells (PSCs) have greatly improved our understanding of human brain development and disease. Among these, 3D cortical organoid systems are able to recapitulate some aspects of in vivo cytoarchitecture of the developing cortex. Here, we tested three cortical organoid protocols for brain regional identity, cell type specificity and neuronal maturation. Overall, all protocols gave rise to organoids that displayed a time-dependent expression of neuronal maturation genes such as those involved in the establishment of synapses and neuronal function. Comparatively, guided differentiation methods without WNT activation generated the highest degree of cortical regional identity, whereas default conditions produced the broadest range of cell types such as neurons, astrocytes and hematopoietic-lineage-derived microglia cells. These results suggest that cortical organoid models produce diverse outcomes of brain regional identity and cell type specificity and emphasize the importance of selecting the correct model for the right application.

## 1. Introduction

The study of the human cortex during development is critical for a better understanding of how this rostral brain region evolves, including its neuronal connectivity and function as well as how it is affected in a range of human disorders. The ability to isolate and grow mouse and human embryonic stem cells (ESCs) and the discovery of directed and undirected ways to generate neural cells have allowed the generation of a range of human brain cellular models [1,2,3]. An important milestone in this respect was the original description of the derivation of mouse cortical cells from mouse ESCs in the absence of any morphogens. This protocol constituted the first proof-of-principle that default non-directed methods for neural differentiation generate mostly cells of the rostral neural tube, such as the cortex [4,5,6]. Similarly, human ESCs were later differentiated following default conditions to generate cortical progenitors and neurons [7,8]. These findings showed that the absence of caudalizing factors was sufficient to induce cortical identity in vitro. In an effort to increase the number of cortical cells derived from human embryonic stem cells (hESCs) and human induced pluripotent stem cells (hiPSs), different conditions such as the addition of retinoic acid to enhance neuronal maturation, or the use of dual SMAD inhibitors to block TGFβ and BMP signaling pathways, have been used to enhance neuroectodermal fate [9,10,11]. Subsequent addition of morphogens such as WNT, BMP, FGF8 and SHH was shown to pattern areal identity within the telencephalon [4,7,12,13,14,15] or to caudalize areal identity towards posterior midbrain and hindbrain fates [16,17]. These in vitro models recapitulated important aspects of in vivo brain development such as temporal patterning and human-specific patterns of neuronal maturation [5,7,11]. Although these two-dimensional (2D) telencephalic models proved to be invaluable in studying important aspects of human brain evolution and diseases [4,7,14,15,18,19,20,21,22,23], other aspects such as long-term cell polarity as well as neuronal migration and integration into neural networks could not be as easily investigated in these reductionist in vitro systems. In one approach, the developmental potential of mouse and human ESC-derived cortical cells has been tested after transplantation into the more suitable environment of the mouse brain in vivo [5,7,15,24,25,26,27]. In another approach, in vitro tridimensional (3D) models of corticogenesis have been developed from human PSCs that recapitulate to some extent in vivo cortical cytoarchitecture and cell polarity in a dish. Such models have provided a cellular environment that supports the formation of ventricle-like structures around which progenitors lie within a ventricular-like region (VZ) separated from a rudimentary cortical plate-like area [28,29,30,31,32]. Three-dimensional organoid models are either based on default differentiation, where human neuroectodermal cells acquire a rostral forebrain identity in the absence of caudalizing signals [28,33,34], or based on the use of morphogenes that direct the differentiation towards specific neuronal subtypes [29,30,31]. Guided differentiation protocols incorporate a combination of different morphogens such as the use of dual SMAD inhibition to enhance neuroectodermal fate [9,29,31]. This first “neural induction” step is followed by an amplification step or proliferative step in the presence of EGF and FGF [29] or in the presence of a TGFβ inhibitor together with a GSK3β inhibitor, which results in activation of WNT signaling [31]. Cortical organoid protocols may also use morphogens to enhance the maturation profiles; embed organoids in a basement membrane-like matrix to support polarity and cell growth; and/or use bioreactors to increase oxygen and nutrient diffusion into the tissue structures at late stages in the differentiation [29,30,31,35]. However, to date, it remains unclear which conditions are necessary and sufficient to generate specific brain cell types, to best recapitulate in vivo cell polarity and to generate mature neuronal populations.

Here, we explored the differentiation capacity of three cortical 3D organoid models, one of them based on default differentiation and on the use of a matrix basement [28] and the other two being guided differentiation protocols with or without matrix inclusion of the organoids in the absence or presence of WNT activation [29,31]. We analyzed the organoids at the early progenitor and late neuronal stages and identified their brain regional identity based on the expression of regional-specific genes from the rostro-caudal axis of the neural tube. Furthermore, we screened the organoids for the expression of neuronal maturation genes and compared them with human embryonic and adult cortical brain samples. Finally, we compared the various models for their ability to generate different brain cell types such as oligodendrocytes, astrocytes and non-neuroectodermal derivatives such as microglia cells.

## 2. Results

### 2.1. Default and Guided Differentiation Cortical Organoid Models Show a High Efficiency of Neuronal Specification

We differentiated hESC-H9 cells using a default “intrinsic protocol” in the absence of any morphogenes [28] (PC1), a directed differentiation protocol following dual SMAD inhibition and GSK3β inhibition/WNT activation coupled with TGFβ inhibition [31] (PC2) or a directed differentiation protocol with dual SMAD inhibition followed by EGF and FGF treatment [29] (PC3) (Figure 1A). H9 cells were aggregated and exposed to default or directed differentiation followed by (i) a Matrigel-embedding step from 2 weeks onwards (PC1), (ii) a Matrigel-embedding step for about 1 week (PC2) or (iii) no Matrigel-embedding step (PC3) (Figure 1A). These protocols were adapted to the use of an orbital shaker from 2 weeks in vitro until the last time point in the culture to promote the diffusion of oxygen and nutrients inside the organoids, as an alternative to the use of a spinning bioreactor (see Section 4) [36]. After the neuroectodermal induction step, organoids were transferred to neuronal medium and left to mature until 3 months in culture without (PC1), or in the presence of neuronal maturation morphogens such as BDNF and GDNF or NT3 (PC2 and PC3) to accelerate and improve the outcome of mature neuronal populations (Figure 1A). We analyzed organoids from PC1, PC2 and PC3 protocols at early neuronal progenitor (days 7, 25) and late neuronal stages (days 50, 90). As expected, the size of the organoids increased over time (Appendix A).

In order to better understand the differences between organoids derived from the three protocols, we performed a high–throughput quantitative PCR analysis by Biomark HD assay (Fluidigm) (see Appendix A). Organoids derived from the PC1 method showed the highest variability based on principal component analysis (PCA) (Appendix A), suggestive of a high heterogeneity in cell fate identity by default methods, as reported [28,35,37].

Stem cell renewal genes were, as expected, downregulated following differentiation (Appendix A). Interestingly, mesoderm germ layer markers such as CDX2 and KRT14 were enriched in PC1- and PC2-derived organoids, indicating the presence of a pool of non-neuronal cell types in these conditions [38] (Appendix A). The three protocols generated organoids that expressed proliferative and neuroectodermal progenitor markers such as nestin, KI67, PAX6, SOX2 and BLBP. The percentage of neuronal progenitors and proliferative cells decreased with time (Figure 1B–G, and Appendix A), suggesting an early phase of progenitor expansion (day 25) followed by a differentiation mature phase (day 50). We detected a higher presence of PAX6+ cells at the early stage in PC2-derived organoids, suggestive of more efficient and/or faster neuroectoderm acquisition following the longer exposure to dual SMAD/TGFβ inhibition (from day 0 to day 14) (Figure 1F,G), as reported [9]. Next, we analyzed the pattern of expression of rostral and caudal neural tube markers (Figure 1H). We observed a time-dependent enrichment in rostral regional markers in organoids from all three protocols (Figure 1I and Appendix A). In contrast, caudal brain regional markers such as GBX2, LMX1A, HOXA2 and HOXA4 were less abundant but more represented in PC1 and PC2 organoids (Figure 1J and Appendix A). In addition, the neuronal markers doublecortin (DCX) and β3-tubulin (β3TUBB) were broadly expressed in organoids from all protocols (Appendix A).

All these data indicate that the three protocols generate organoids with an overall expression of neuronal progenitor and rostral neural identity markers as well as neuronal genes suggesting an efficient neuroectodermal acquisition and neurogenesis in all conditions, as expected.

### 2.2. 3D Organoid Differentiation Protocols Closely Mimic In Vivo Cell Polarity and Cortical Cytoarchitecture

In vitro organoid models of corticogenesis from hESCs recapitulate the presence of polarized structures with an apical and basal side resembling the in vivo ventricular zone (VZ) of the developing cortex, called VZ-like regions. However, to date, it is not clear to what extent the different cortical organoid models show differences in cell polarity. In order to compare cell polarity among protocols, we analyzed day 25 organoids for the localization of progenitor markers in apical vs. basal compartments and day 90 organoids for the separation of progenitor and neuronal markers. We first analyzed the localization of RG mitotic cells within VZ-like regions. We detected mitotic phospho-vimentin-positive RG cells close to PALS1 + apical side of the VZ-like regions in organoids derived from all of the three protocols (white arrows; Figure 2A) mimicking in vivo cell polarity [28,32]. Interestingly, the area of VZ-like regions was larger in PC1-derived organoids on day 25 (Figure 2A–D,F,G). Accordingly, the radial orientation of the RG nestin-positive scaffold appeared most organized in PC1-derived organoids with the highest localization of KI67+ and SOX2+ progenitor cells inside VZ-like regions (Figure 2B–D,H,I).

These data show that all cortical organoid models mimic in vivo cell polarity and overall cytoarchitecture of the developing embryonic cortex where PC1 organoids display the most conserved cell polarity in vitro following long-term Matrigel-embedding and default non-directed conditions. These differences among protocols could suggest an effect of Matrigel, as previously suggested [28,30,36], and/or the presence of neural induction morphogenes on the growth of neuroepithelia and cell polarity.

### 2.3. Long-Term Culture of Cortical Organoids Leads to the Differentiation of Mature Neuronal Populations

Human neurons derived from ESCs and iPSCs have been shown to follow a human species-specific program for maturation characterized by a very slow acquisition of mature neuronal cortical features, whether in 2D in vitro paradigms or following transplantation into the developing mouse brain in vivo [7,10,11,26]. In order to compare the level of neuronal maturation among protocols, we assessed the expression of several mature neuronal markers in organoids through time. We observed broad expression of neuronal markers on days 50 and 90, with a time-dependent increase in the number of NEUN+ and MAP2+ neurons in organoids derived from all protocols (Figure 3A–D and Appendix A). The expression levels of genes involved in neuronal maturation and genes involved in the establishment of synapses were similarly upregulated in a time-dependent fashion (Figure 3E–I). We noted few differences among protocols, such as lower expression of PSD95 and GRIN2B in PC3-derived organoids compared to PC2 and lower expression of GRIA1 in PC2-derived organoids. Whether this could be the result of small differences in neuronal maturation resulting from the different medium composition/morphogen conditions, we could also not rule out that these molecular differences are inherent to different brain areal identities obtained from the different protocols (Figure 3E–I). Next, we compared the level of expression of the adult mature splice form of the MAPT gene (TAU), 4R TAU [39], in three-months organoids and human embryonic and adult brain samples. We confirmed that the pattern of expression of 4R TAU varies between embryonic and adult stages in humans (Figure 3J) [24,40]. The expression of 4R TAU in three-month cortical organoids was similar to that in the human embryonic cortex (Figure 3J). Moreover, cluster heatmap representation of neuronal maturation genes revealed that late (day 90) brain organoid samples from PC1, PC2 and PC3 more closely resemble embryonic cortical samples (9PCW; 20PCW) than early (day 25) brain organoids or human ESCs (day 0) (Figure 3K–M). Accordingly, the top 30 Biomark genes from three-month organoids revealed fetal brain identity and protocol-specific categories (PC1: prefrontal cortex and spinal cord; PC2: prefrontal cortex and midbrain; PC3: prefrontal cortex and cerebral cortex) (Appendix A).

These results suggest that brain organoids show a time-dependent maturation with increasing expression of genes essential for neuronal function such as subunits of AMPA and NMDA receptors and postsynaptic density proteins.

### 2.4. Directed Differentiation without WNT Agonists Generates Organoids with the Highest Telencephalic/Cortical Identity

To understand the differences in brain regional identity of 3D cortical organoid models, we compared default and guided conditions for cortical excitatory and inhibitory neuronal identity progenies. Strikingly, we found a higher proportion of TBR1+ deep layer cortical neurons in PC3-derived organoids and higher proportion of CTIP2+ (deep layers) and SATB2+ (upper layers) neurons in PC1- and PC3-derived organoids (Figure 4A–C,G). Moreover, we found that PC3-derived organoids showed higher levels of the glutamatergic marker VGLUT1 and higher VGLUT1 puncta density on day 90, suggestive of a higher glutamatergic identity (Figure 4D,H and Appendix A). Interestingly, we also detected a higher percentage of GAD67+ inhibitory neurons in PC3 organoids (Figure 4E,G). Next, we confirmed these results by Biomark analysis. Telencephalic, cortical and glutamatergic-specific markers such as FOXG1, EMX1, TBR2, NEUROD1, TBR1 and VGLUT1 were more expressed in PC3-derived organoids at three months (day 90), consistent with a higher degree of telencephalic, cortical and glutamatergic identity of PC3-derived organoids (Figure 4I,J). Similarly, overall markers of inhibitory GABAergic neurons of the cortex such as GAD1, GAD2 and CALB1 were expressed at a higher level in PC3-derived organoids (Figure 4K), suggestive of a higher presence of interneurons in PC3 organoids. Moreover, representation of the top 10 upregulated Biomark genes from each protocol using the Human Brain Atlas database showed a comparative enrichment of human cerebral cortex and basal ganglia-specific genes for PC3 organoids, and a broader regional identity for PC1 default differentiation and PC2 guided condition with WNT activation (Appendix A).

These data suggest that guided differentiation without WNT activation shows the highest enrichment of cortical, glutamatergic and GABAergic inhibitory neuronal progenies, suggestive of higher rostral cortical identity following dual SMAD inhibition and in the absence of caudalizing factors such as WNT activation.

### 2.5. Default Differentiation Generates Brain Organoids with a Broad Range of Glial Cell Subtypes

Models for cortical organoid differentiation have been suggested to generate glia derivatives such as astrocytes, oligodendrocytes, and few mesodermal hematopoietic lineage-derived cells such as microglia cells [28,29,31,38,41]. However, to date it is unclear to which extent these cell types are generated and enriched in different protocols. In order to characterize the presence of these cell types we analyzed the presence of cell-specific subtype markers within organoids. We found increased expression of astrocyte and microglia markers in PC1-derived organoids (Figure 5A–F and Appendix A), suggesting a higher presence of these glia cell types following default differentiation conditions. In contrast, even though some oligodendrocyte markers were detected in late stage organoids we could not identify the presence of double-positive OLIG2+/NKX2.2+ late oligodendrocyte progenitor cells (OPC) in any of the conditions tested (Appendix A).

A comparison of the top 30 Biomark genes from three-month organoids derived from the three protocols revealed that 45% of genes were common, 7% of genes were specific for PC1 (astrocyte-specific and caudal SNC identity), 7% of genes were specific for PC2 (neuronal maturation and caudal identity) and 17% of genes were specific for PC3 (telencephalic and cortical identity) (Figure 5G).

Next, we analyzed the 20 most differentially expressed genes between late (three months) and early brain organoids (day 25) to assess overall differentiation capacity among protocols. Interestingly, the PC1 organoids’ top 10 upregulated genes at three months showed an upregulation of astrocyte and microglia genes (Figure 5H) on day 90 compared to day 25. The PC2 organoids’ top 10 upregulated genes were in the category of astrocyte-specific as well as caudal neural identity, whereas rostral and cortical-specific genes were downregulated (Figure 5I). We found that cortical-specific, ventral telencephalic and neuronal maturation genes were among the top 10 upregulated genes in three-month PC3 organoids. Conversely, neuronal progenitor, caudal identity and proliferative genes were among the top 10 downregulated (Figure 5J). Accordingly, when comparing the top 10 upregulated genes from each protocol across all organoids, we observed that cortical-specific (red), cortical layer subtype (pink), rostral identity (yellow) and GABAergic identity (green) gene categories were best represented in PC3 late organoids (Figure 5K).

These data suggest that directed differentiation without WNT activation generates the highest enrichment of rostral telencephalic, both cortical glutamatergic and ventral GABAergic, cell identities. Spontaneous differentiation, on the other hand, leads to the highest number of glial cell subtypes in vitro.

## 3. Discussion

Several 3D cortical organoid protocols have been published to date [28,29,30,31,42]. Among those, default differentiation from human PSCs has proven to be sufficient to generate cortical-like structures in the absence of any morphogens [28,33,34]. Other protocols rely on an initial dual SMAD inhibition step to enhance neuroectodermal cell fate followed by exposure to growth factors to promote neuronal maturation [29,30,31,42]. In addition, some protocols include the embedding of human 3D brain structures into a basement membrane-like extracellular matrix only for one week at an early stage [31], or from two weeks onwards [28,43], whereas others have grown organoids without these scaffolds [29,30,42].

Here, we compared three methods for cortical organoid generation from human ESCs based on directed or default differentiation strategies, with or without matrix scaffold, and we have analyzed cell progeny identities in culture. We found that methods based on directed differentiation without WNT activation [29,44] generated the highest level of cortical glutamatergic and inhibitory GABAergic cell identity, corresponding to a higher rostral forebrain/telencephalic identity. Furthermore, we found that the addition of GSK3β inhibitors/WNT signaling pathway activators in a guided differentiation protocol generates more caudal neural tube cell identities [31], as previously suggested in 2D models [16,17]. Default differentiation conditions [28,43] generated organoids composed of the broadest range of brain areal identities and brain cell subtypes, and showed the highest variability between organoids, as reported [28,35,37].

Conditions where cell polarity is best displayed and preserved at early and late time points in culture are essential factors to consider when choosing in vitro models suitable for the study of cell division, and neuronal migration defects in developmental diseases [28,45,46]. To that end, we compared the three protocols for the degree of polarity displayed in culture. We found a separation between progenitor regions and the localization of differentiated neurons in all organoid protocols. However, organoids embedded into Matrigel generated the best organization of neuronal progenitors around ventricle-like structures with an extended radially oriented scaffold from radial glia cells. Furthermore, organoids embedded in Matrigel scaffolds from about two weeks until the latest time point in culture resulted in the most elongated VZ-like neuroepithelium and polarized regions within organoids. Thus, these methods could be more suitable for the study of cell polarity of human cortical progenitors in normal development and in neurodevelopmental diseases.

An important feature for human cortical in vitro models is the degree of neuronal maturation reached in culture. Previous studies using 2D models have shown that human ESC-derived cortical neurons display a slow maturation program highly reminiscent of the human species. Indeed, 2–3-month-old cultures have shown an expression profile comparable to that of the human fetal brain at midgestational stages [7,10,11]. Studies comparing 3D and 2D paradigms have suggested a slightly faster maturation profile of brain-like structures following organoid models [29,47]. However, single-cell RNAseq analysis revealed that the cellular composition of 3D organoids after 3 and even 6 months in culture still appears to be similar to the human fetal brain rather than postnatal or adult stages [35,37]. More recently, protocols have used modified conditions, such as the use of bioreactors or slicing the organoids, in an attempt to improve neuronal maturation and survival of neural populations after months in vitro [30,31,34,35]. Despite all this, the associated in vivo age of organoids has remained similar to gestational weeks 19–24 of the developing human fetal brain [29,31,34,35,37,48]. We compared the neuronal maturation profile across time in organoids derived from the three protocols. We observed that protocols for default differentiation or directed differentiation resulted in a time-dependent increase in the expression of neuronal maturation genes such as NEUN and the MAP2 structural gene; genes important for the formation and maintenance of synapses such as PSD95; genes involved in synaptic plasticity such as CAMKII; and genes involved in neuronal activity such as GRIA1, GRIN2B and vGLUT1. We speculate, however, that there may be differences in the level of functional activity among brain organoids derived from the three protocols tested, linked to the ratio of neurons to glial subtypes, whose role is essential for the establishment and maturation of synapses. The present study was based on the use of molecular and cellular observations to determine neuronal maturation, and future studies are required to fully test and compare the neuronal functional activity of brain organoids generated by different protocol methods.

Further, we looked into the expression of a specific splice form of the MAPT gene enriched in the adult brain, the 4R TAU [24], which is associated with different tauopathies such as frontotemporal dementia and Alzheimer’s disease [49]. We found a similar expression among organoids from different conditions, which matched that found in the human embryonic, rather than adult, cortex. All of these findings support a late embryonic identity of the organoid samples after three months in culture. It would be interesting in future works to explore the conditions that could further accelerate aging in 3D organoids in vitro such as the use of factors that inhibit telomerases; the expression of truncated proteins from patients showing accelerated aging, such as progerin from Hutchinson–Gilford progeria syndrome (HGPS); the inhibition of autophagosome function; and the induction of DNA damage [50,51,52].

A general aim of 3D organoid models vs. reductionist 2D paradigms is to generate a more physiological cellular environment that can more closely recapitulate essential aspects of normal in vivo cortical development. Cortical development is a fine-tuned process where different cellular types interact with each other and balance cell proliferation, cell cycle exit, neuronal migration and neuronal maturation [53]. In this aspect, it would be interesting to define the experimental conditions that more closely generate the broad spectrum of cell identities present in the cortex at different developmental stages. Most models of cortical organoid formation have shown the presence of additional brain cell types such as astrocytes, and some degree of oligodendrocytes and non-neuroectodermal-derived cells such as hematopoietic-lineage-derived microglia cells of the brain [28,29,31,38,41]. However, due to the diversity of protocols and experimental conditions, it is currently unclear which paradigms lead to a higher enrichment of these particular cell populations. We noticed the absence of late oligodendrocyte precursor cells and mature oligodendrocytes in organoids derived from all protocols, even at the latest time points tested. These results could indicate that longer periods in culture beyond three months are necessary to generate oligodendrocyte populations. This observation could reflect an intrinsic timing of oligodendrocyte generation related to postnatal stages in vivo [37]. An alternative explanation is that some factors essential for oligodendrocyte generation or maturation are lacking in the in vitro conditions. Of interest in this respect are recent studies using modified protocols for the generation and enrichment of oligodendroglia cell populations through the exposure of neuroectodermal progenitors to EGF, FGF2, SHH agonists and oligodendroglia trophic factors that have been shown to be required for their maturation [41,54]. Interestingly, intrinsic default differentiation showed the highest yield of glial cell types such as astrocytes and hematopoietic-lineage-derived microglia cells, in line with what was seen histologically [28,35,37,38]. This suggests that default differentiation methods could be essential to recapitulate cell diversity in a dish to best study the brain in health and disease.

In summary, we show that cortical organoid models generate a wide variety of cell identities corresponding to different brain areal identities and brain cell types that relate to the use of default or directed differentiation conditions. Moreover, we detected changes in cell polarity and/or size of VZ-like regions that could be related to the presence or absence of extracellular matrix scaffolds, as previously suggested [28,30,36], and/or neural induction morphogens. Our results also highlight that cortical organoids in culture display a time-dependent neuronal maturation that nevertheless remains similar to that from the embryonic cortex, rather than to adult cortical samples, even after three months. Our findings point out the importance of selecting the right cortical organoid differentiation method according to the areal identity, brain cell subtype and cell polarity degree required for each specific brain model.

## 4. Methods and Materials

### 4.1. Human ESC Differentiation into 3D Cortical Organoid Models

Cortical organoids were generated with few modifications (see Appendix A and methods) from previously published methods (PC1: [28]; PC2: [31]; PC3: [29]).

### 4.2. Quantification of Percentage of Cells, Mean Intensity and Number of Puncta per Area

Stained organoid sections were imaged by confocal microscopy (TSC SP5, Leica Microsystems) with a 20× objective to image Z series stacks of areas inside human organoids. All images were acquired using identical acquisition parameters as 16-bit, 1024 × 1024 arrays. Percentage of cells, mean intensity and number of puncta per area were quantified using Fiji/ImageJ (version 2.1.0/1.53c National Institutes of Health, Bethesda, MD, USA) software.

Percentage of cells (%) and number of puncta were calculated using a default threshold and generating masks to identify particles. Number of puncta per area was calculated by dividing the number of identified puncta by the area. Mean intensity of IF images was subtracted from background values. Number of puncta per area and mean intensity values were related to an internal control value of 1. Data are represented as mean values. Statistical analysis was performed with two-way ANOVA with Tukey post-tests.

### 4.3. Quantification of Organoid Size

Organoid size was analyzed on days 25, 50 and 90 for in vitro organoids. Fixed, cryopreserved and cryosectioned organoids (12–15 μm sections) were imaged following nuclear staining using Hoechst using a Leica Confocal SP5 microscope (Leica Microsystems, Wetzlar, Germany) with an air 4× objective driven by Las-AF software. Image J/Fiji software was used to evaluate the area of the organoid pictures and represented as mm^2^.

For day 25 and day 90 PC1 (*n* = 4 organoids), PC2 (*n* = 4 organoids) and PC3 (*n* = 4 organoids), statistical analysis was performed using a two-way ANOVA with Tukey post-tests; ns = non-significant.

### 4.4. Human Embryonic and Adult Cortical Samples

Post-mortem human fetal cortical specimens from 9 pcw were obtained from the University of Cambridge, UK. All procedures were approved by the research ethics committees and research services division of the University of Cambridge and Addenbrooke’s Hospital in Cambridge in accordance with the Human Tissue Act 2006. RNA from post-mortem human brain specimens at 20 pcw was obtained according to an established protocol [55]. Post-mortem adult cortical samples were obtained from the Harvard Brain Tissue Resource Center (HBTRC) (Belmont, MA, USA). Tissue was handled in accordance with ethical guidelines and regulations for the research use of human brain tissue set forth by the National Institute of Health (NIH) accessed on 1 January 2019 (http://bioethics.od.nih.gov/humantissue.html, accessed on 1 January 2019) and the World Medical Association Declaration of Helsinki (https://www.wma.net/policies-post/wma-declaration-of-helsinki-ethical-principles-for-medicalresearch-involving-human-subjects/, accessed on 1 January 2019). 

### 4.5. Biomark HD Assay (Fluidigm)

For high-content qPCR experiments, cDNA was pre-amplified using a 0.2× pool of primers prepared from the same gene expression assays as were used for the Biomark analysis. Gene expression experiments were performed using the 96 × 96 qPCR Dynamic Array microfluidic chips (Fluidigm). Housekeeping genes used for normalization of the data were YWHAZ and GADPH. Data were analyzed and gene expression was carried out through the use of ΔCT as normalized absolute gene expression analysis: 2^−ΔCt^ whereby ΔCt = Ct_target_ − Ct_normalizer_. Statistical analyses were performed using a one-way analysis of variance (ANOVA) test, with Tukey post-tests, and indicated as * for *p* < 0.05, ** for *p* < 0.01, *** for *p* < 0.001 and **** for *p* < 0.0001.

### 4.6. Statistical Analysis

Data from experiments were processed with GraphPad Prism v.7 software. Statistical analysis was performed using a one-way analysis of variance (ANOVA) or two-way ANOVA test, with Tukey post-tests, and indicated as * for *p* < 0.05, ** for *p* < 0.01, *** for *p* < 0.001 and **** for *p* < 0.0001.

## Figures and Tables

**Figure 1 ijms-23-13159-f001:**
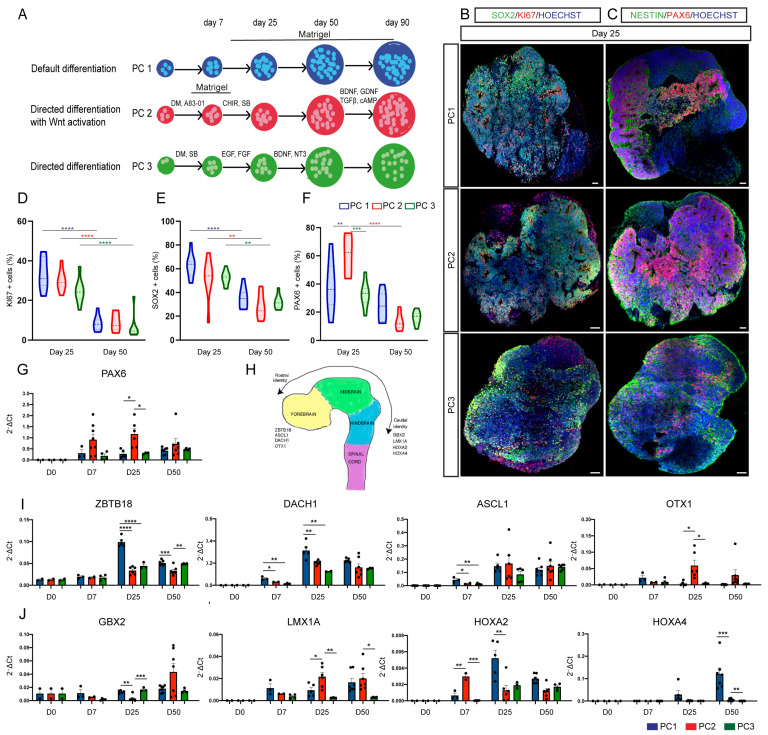
Default and directed differentiation protocols generate organoids that are enriched in rostral neural progenitors. (**A**) Schematic representation of the experimental outlay. PC1 (default differentiation); PC2 (directed differentiation with WNT activation); PC3 (directed differentiation without WNT activation). Organoids were collected at early progenitor stages (D7, D25) and late neuron stages (D50, D90). (**B**,**C**) Immunofluorescence images showing the expression of the neuronal progenitor markers SOX2 (green; (**B**)) and nestin (green; (**C**)), the proliferation marker KI67 (red; (**B**)), and the neuroectoderm progenitor marker PAX6 (red; (**C**)) on days 25 in organoids derived from PC1, PC2 and PC3 protocols. Scale bars represent 100 μm (**B**,**C**). (**D**–**F**) Violin plots showing the percentage of cells expressing KI67 (**D**), SOX2 (**E**) and PAX6 (**F**) in PC1 (blue), PC2 (red) and PC3 (green) organoids. *n* = 8 sections from 4 organoids per protocol. Two-way ANOVA (**D**–**F**) with Tukey post-tests. * *p* < 0.05, ** *p* < 0.01, *** *p* < 0.001, **** *p* < 0.0001. (**G**,**I**,**J**) Biomark analysis of H9 cells (day 0 *n* = 2) and PC1 (blue bars), PC2 (red bars) and PC3 (green bars) organoids (days 7, 25, 50) for the expression of the dorsal forebrain-specific gene PAX6; the rostral markers ZBTB18, DACH1, ASCL1 and OTX1; and the caudal identity genes GBX2, LMX1A, HOXA2 and HOXA4. Data are shown as absolute normalized amount of mRNA (2^−^^Δ^^Ct^) ± SEM; PC1 (*n* = 3–5), PC2 (*n* = 2–7), PC3 (*n* = 3–4). Two-way ANOVA with Tukey post-tests. * *p* < 0.05, ** *p* < 0.01, *** *p* < 0.001, **** *p* < 0.0001. (**H**) Cartoon depicting the rostro-caudal axis of the developing neural tube.

**Figure 2 ijms-23-13159-f002:**
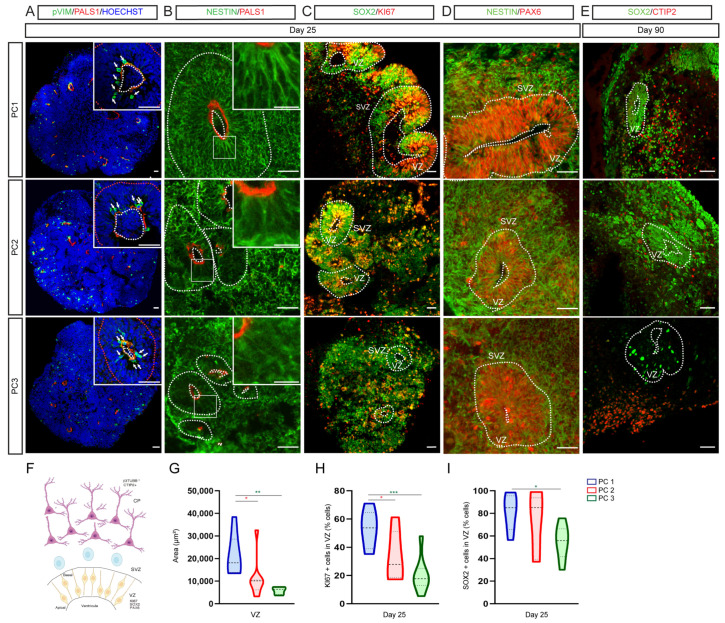
Protocols for cortical organoid differentiation recapitulate important aspects of cortical in vivo polarity. (**A**–**D**) Immunofluorescence images showing the expression of the apical marker PALS1 (red; (**A**,**B**)), the mitotic marker p-vimentin (green; (**A**)), the neuronal progenitor markers nestin (green; (**B**,**D**)) and SOX2 (green; (**C**), the proliferative marker KI67 (red; **C**) and the cortical progenitor marker PAX6 (red; (**D**)) on day 25. (**E**) Immunofluorescence images showing the expression of the neuronal progenitor marker SOX2 (green) and the deep layer neuronal marker CTIP2 (red) on day 50. CTIP2-positive neuronal cells are found outside the VZ-like progenitor region. Counterstaining of nuclei was performed with Hoechst (blue). Dashed lines delineate the VZ-like proliferative regions in the organoids and the apical region of the VZ. Arrows show the presence of numerous p-vimentin mitotic cells located at the apical side of the PALS1+ VZ (**A**). VZ, ventricular zone; SVZ, subventricular zone. Scale bars represent 100 μm (**A**) and 50 μm (**B**–**E**), and those in the top right insets represent 400 μm (**A**) and 100 μm (**B**). (**F**) Cartoon depicting the apical–basal localization of radial glia cells within the VZ and the progenitor and neuronal cytoarchitecture of the developing telencephalon. (**G**) Violin plot showing the area (μm^2^) of VZ-like areas in PC1 (blue), PC2 (red) and PC3 (green) organoids. (**H**,**I**) Violin plots showing the percentage of cells expressing KI67 (**H**) and SOX2 (**I**) inside VZ-like regions from PC1 (blue), PC2 (red) and PC3 (green) organoids. *n* = 8 sections from 4 organoids per protocol. One-way ANOVA with Tukey post-tests. * *p* < 0.05, ** *p* < 0.01, *** *p* < 0.001.

**Figure 3 ijms-23-13159-f003:**
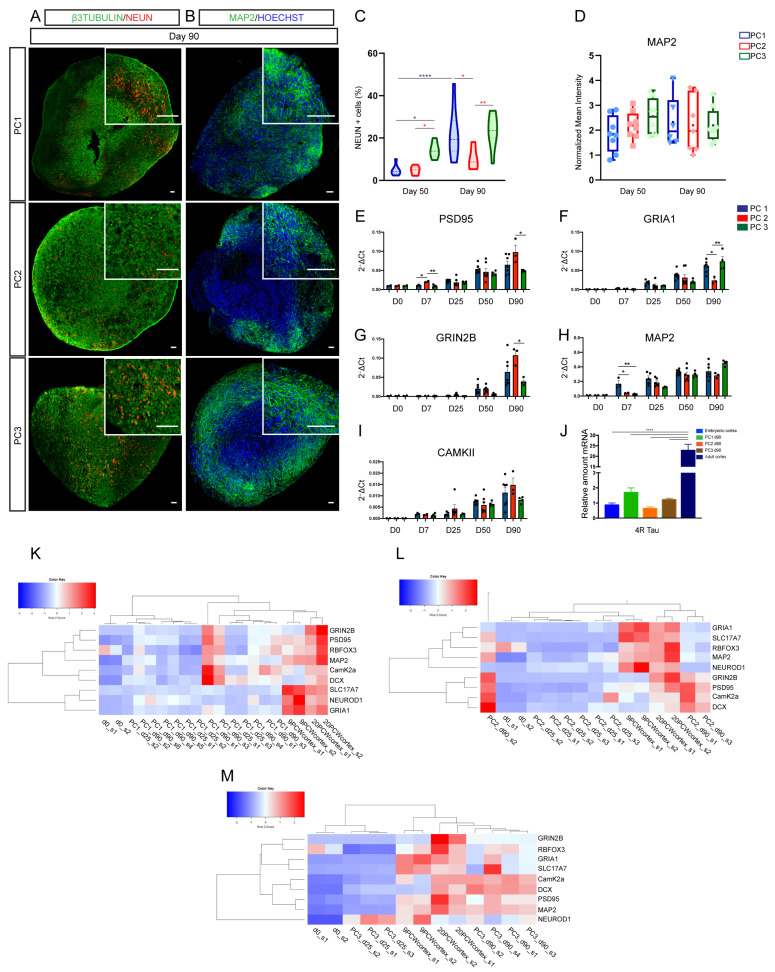
Long-term culture of 3D organoids leads to the generation of mature neuronal populations. (**A**,**B**) Immunofluorescence images showing the expression of the neuronal and maturation markers β3-tubulin (green; (**A**)), NEUN (red; (**A**)) and MAP2 (green; (**B**)) in day 90 organoids from PC1, PC2 and PC3. Counterstaining of nuclei was performed with Hoechst (blue). Scale bars represent 100 μm, and those in the top right insets represent 400 μm. (**C**,**D**) Violin and box graph plots showing the percentage of cells expressing NEUN (**C**) and the normalized mean intensity of MAP2 (**D**) in PC1 (blue), PC2 (red) and PC3 (green) organoids. *n* = 8 sections from 4 organoids per protocol. Two-way ANOVA (**C**,**D**) with Tukey post-tests. * *p* < 0.05, ** *p* < 0.01, **** *p* < 0.0001. (**E**–**I**) Biomark analysis of H9 cells (day 0 *n* = 2) and PC1 (blue bars), PC2 (red bars) and PC3 (green bars) organoids (days 7, 25, 50, 90) for the expression of neuronal maturation genes: PSD95, GRIA1, GRIN2B, MAP2 and CAMKII. Data are shown as absolute normalized amount of mRNA (2^−Δ^^Ct^) ± SEM; PC1 (*n* = 3–7), PC2 (*n* = 2–7), PC3 (*n* = 3–4). Two-way ANOVA with Tukey post-tests. * *p* < 0.05, ** *p* < 0.01. (**J**) qRT-PCR analysis of PC1 (blue bars), PC2 (red bars) and PC3 (green bars) organoids (day 90); embryonic cortex; and adult cortex for 4R TAU (MAPT) isoform expression. Data are shown as relative amount of mRNA compared to embryonic cortex, as value 1 ± SEM (fold change). PC1, PC2 and PC3 *n* = 3; embryonic cortex PCW9 *n* = 2; and adult cortex (52–63 years) *n* = 2. One-way ANOVA with Tukey post-test for all samples, **** *p* < 0.0001. (**K**–**M**) Cluster heatmap analysis showing the normalized expression of neuronal maturation genes in early (day 25) and late (day 90) PC1, PC2 and PC3 organoids; two samples of undifferentiated H9 cells (day 0: s1, s2); and two samples from human embryonic cortex at 9 and 20 gestational weeks (9/20 GW) (s1, s2). Top: legend showing color code for comparative levels of expression.

**Figure 4 ijms-23-13159-f004:**
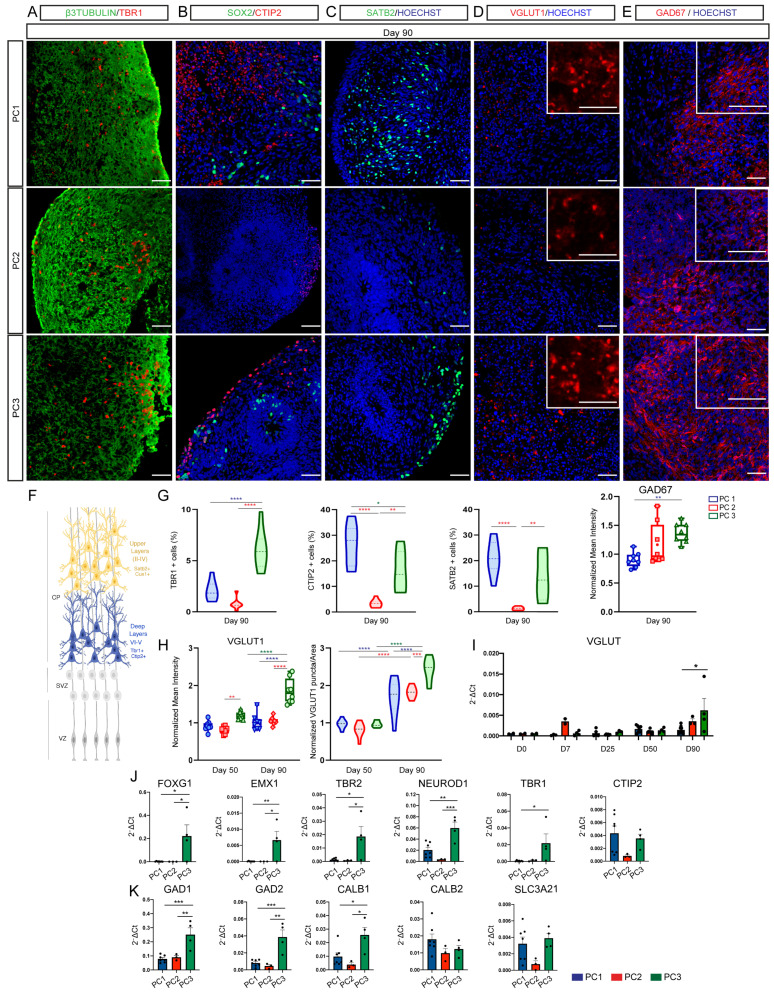
Directed differentiation without WNT agonists generates the highest cortical-specific progenies. (**A**–**E**) Immunofluorescence images showing the expression of the neuronal marker β3-tubulin (green; (**A**)), the neuronal progenitor marker SOX2 (green; (**B**)), the deep layer neuronal markers TBR1 (red; (**A**)) and CTIP2 (red; (**B**)), the upper layer neuronal marker SATB2 (green; (**C**)), the glutamatergic marker VGLUT1 (red; (**D**)) and the GABAergic marker GAD67 (red; (**E**)) in organoids following PC1, PC2 and PC3 protocols. Counterstaining of nuclei was performed with Hoechst (blue). Scale bars represent 50 μm (**A**–**C**,**E**) and 100 μm (**D**), and those in the top right insets represent 400 μm (**D**) and 100 μm (**E**). (**F**) Cartoon depicting the different progenitor regions (VZ, SVZ) and neuronal compartments (CP: deep layers; upper layers) of the developing cortex. (**G**) Violin and box graph plots showing the percentage of cells expressing the cortical layer markers TBR1, CTIP2 and SATB2 and the normalized intensity of GAD67 in PC1 (blue), PC2 (red) and PC3 (green) organoids. *n* = 8 sections from 4 organoids per protocol. One-way ANOVA with Tukey post-tests. * *p* < 0.05, ** *p* < 0.01, *** *p* < 0.001, **** *p* < 0.0001. (**H**) Box graph and violin plot showing the normalized intensity and puncta density of VGLUT1 in PC1 (blue), PC2 (red) and PC3 (green) organoids. *n* = 8 sections from 4 organoids per protocol. Two-way ANOVA with Tukey post-tests. * *p* < 0.05, ** *p* < 0.01, **** *p* < 0.0001. (**I**) Biomark analysis of H9 cells (day 0 *n* = 2) and PC1 (blue bars), PC2 (red bars) and PC3 (green bars) organoids (days 7, 25, 50, 90) for the expression of the glutamatergic gene VGLUT1. Data are shown as absolute normalized amount of mRNA (2^−^^Δ^^Ct^) ± SEM, PC1 (*n* = 3–7); PC2 (*n* = 2–7); PC3 (*n* = 3–4). Two-way ANOVA (**I**) and one-way ANOVA (**J**,**K**) with Tukey post-tests. * *p* < 0.05. (**J**,**K**) Biomark analysis of PC1 (blue), PC2 (red) and PC3 (green) organoids (day 90) for the expression of telencephalic/cortical-specific and glutamatergic genes: FOXG1, EMX1, TBR2, NEUROD1, TBR1, CTIP2 and GABAergic genes: GAD1, GAD2, CALB1, CALB2, SLC3A21. Data are shown as absolute normalized amount of mRNA (2^−^^Δ^^Ct^) ± SEM, PC1 (*n* = 7); PC2 (*n* = 3); PC3 (*n* = 4). Two-way ANOVA (**I**) and one-way ANOVA (**J**,**K**) with Tukey post-tests. * *p* < 0.05, ** *p* < 0.01, *** *p* < 0.001.

**Figure 5 ijms-23-13159-f005:**
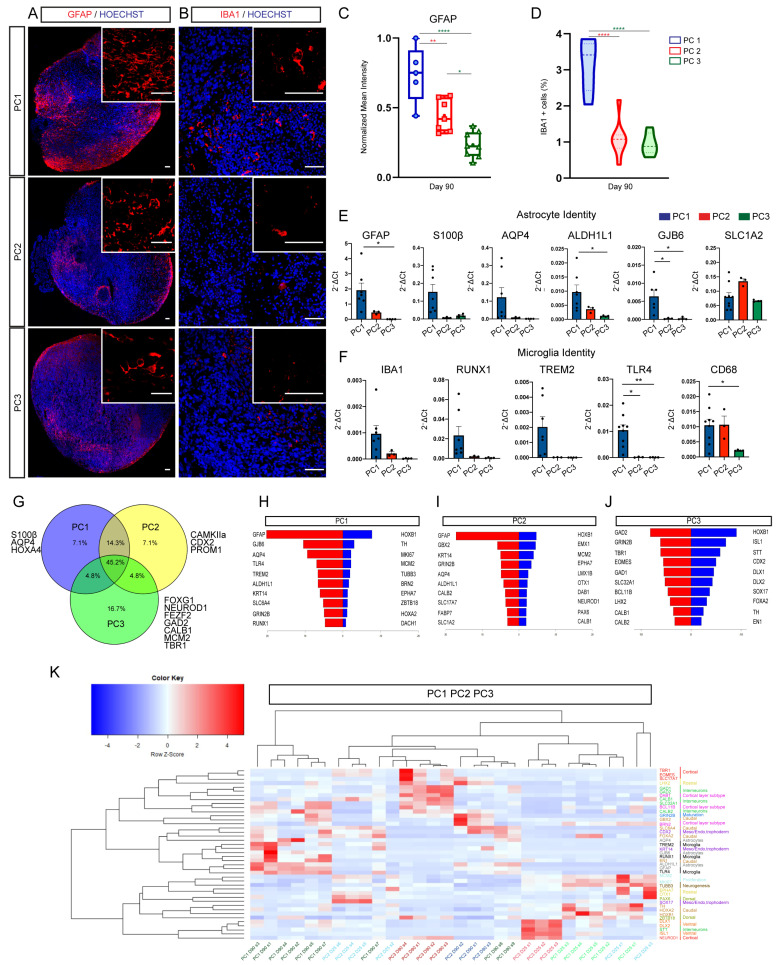
Cortical organoid differentiation protocols generate a diversity of neuronal and non-neuronal populations. (**A**,**B**) Immunofluorescence images showing the expression of the astrocytic marker GFAP (red; **A**) and the microglia marker IBA1 (red; **B**) in organoids following PC1, PC2 and PC3 protocols on day 90. Counterstaining of nuclei was performed with Hoechst (blue). Scale bars represent 100 μm (**A**), 50 μm (**B**) and top right insets 400 μm (**A**) and 100 μm (**B**). (**C**,**D**) Box graph and violin plot showing the normalized mean intensity of GFAP (**C**) and the percentage of cells expressing Iba1 (**D**) in PC1 (in blue), PC2 (in red) and PC3 (in green) organoids. *n* = 8 sections from 4 organoids per protocol. One-way ANOVA with Tukey post-tests. * *p* < 0.05, ** *p* < 0.01, **** *p* < 0.0001. (**E**,**F**) Biomark analysis of PC1 (in blue), PC2 (in red) and PC3 (in green) organoids (day 90) for the expression of astrocytic: GFAP, S100β, AQP4, ALDH1L1, GJB6, SLC1A2; and microglia: IBA1, RUNX1, TREM2, TLR4, CD68 specific genes. Data are shown as absolute normalized amount of mRNA (2^−^^Δ^^Ct^) ± SEM; PC1 (*n* = 7), PC2 (*n* = 3), PC3 (*n* = 4). One-way ANOVA with Tukey post-tests. * *p* < 0.05, ** *p* < 0.01. (**G**) Venn diagram representing the top 30 upregulated Biomark genes from PC1, PC2 and PC3 organoids. (**H**–**J**) Back-to-back representation of the Biomark top 10 upregulated (red, left) and downregulated (blue, right) genes in late (three months) vs. early (day 25) PC1, PC2 and PC3 organoids. (**K**) Heatmap analysis showing the normalized expression of the Biomark top 10 upregulated genes from PC1, PC2 and PC3 in early (day 25) and late (day 90) organoids from all protocols. Gene color code: cortical (red), rostral identity (yellow), interneurons (light green), cortical layer subtype (pink), neuronal maturation (dark blue), caudal identity (light brown), meso/endo/trophectoderm (purple), astrocytes (grey), microglia (black), proliferation (light blue), neurogenesis (dark brown), dorsal (dark green) and ventral genes (orange).

## Data Availability

The datasets generated and analyzed during this study are available from the corresponding authors on reasonable request.

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
