# Peer review of "Brain Regional Identity and Cell Type Specificity Landscape of Human Cortical Organoid Models"

_ijms, 2022, doi:10.3390/ijms232113159_

Round 1

Reviewer 1 Report

The work by Magni et al. compared three organoid protocols displaying cell type specificity and neuronal maturation on each of them. The main aim of the work is to suggest the appropriate model to use according to the brain region of interest. The data are well presented and the images are explanatory.

I therefore suggest a minor revision in order to fix the following point: 

the differences between the three protocols consisted in supplementing the cell medium with different growth factors, in addition to this also the use of matrigel is variable. The authors claim at page 3 that the structural differences among organoids obtained from different protocols "suggest an effect of matrigel on the growth of neuroepithelia and cell polarity with better polarized structures following long term matrigel embedding". Also at page 7 the authors write:"we detected changes in cell polarity... according to the presence or absence of extracellular matrix scaffolds". Since matrigel is not the only variable among the protocols, the above sentences are not scientifically correct. The authors should therefore remove or correct those sentences.

Reviewer 2 Report

In the manuscript titled "Brain regional identity and cell type specificity landscape of human cortical organoid models", the authors tested three cortical organoid protocols, and found that the diverse outcomes from different protocols. Thus, this study emphasizes the importance of selecting the correct model for the right application. This study is well-designed and the results are solid. I only have one question. 

All three protocols led to the generation of mature neuronal population after long-term culture, as shown by the expression of mature neuronal makers and synapse makers. However, markers expression can not tell whether these mature neurons and synapses would have their physiological function. It would be great if the authors could show some functional data (like electrophysiological recording) or at least discuss the potential functions of neurons and synapses from different culture protocols.    
